# A Modified Version of RothC to Model the Direct and Indirect Effects of Rice Straw Mulching on Soil Carbon Dynamics, Calibrated in Two Valencian Citrus Orchards

Simone Pesce [1,2], Enrico Balugani [2,*], José Miguel De Paz [3], Diego Marazza [2] and Fernando Visconti [3,4]

1 Centro Interdipartimentale di Ricerca per le Scienze Ambientali (CIRSA), Alma Mater Studiorum—University of Bologna, 40126 Bologna, Italy; simone.pesce3@unibo.it
2 Department of Physics and Astronomy (DIFA), Alma Mater Studiorum—University of Bologna, Via Zamboni 33, 40126 Bologna, Italy; diego.marazza@unibo.it
3 Instituto Valenciano de Investigaciones Agrarias-IVIA, Centro para el Desarrollo de la Agricultura Sostenible-CDAS, Carretera CV-315, km 10.7, 46113 Moncada, Spain; depaz_jos@gva.es (J.M.D.P.); fernando.visconti@uv.es (F.V.)
4 Departamento de Ecología, Centro de Investigaciones Sobre Desertificación-CIDE (CSIC, UVEG, GVA), Carretera CV-315, km 10.7, 46113 Moncada, Spain
* Correspondence: enrico.balugani2@unibo.it

**Abstract:** The mulching of agricultural soils has been identified as a viable solution to sequester carbon into the soil, increase soil health, and fight desertification. This is why it is a promising solution for carbon farming in Mediterranean areas. Models are used to project the effects of agricultural practices on soil organic carbon in the future for various soil and climatic conditions, and to help policy makers and farmers assess the best way to implement carbon farming strategies. Here, we modified the widely used RothC model to include mulching practices and their direct and indirect effects on soil organic matter input, soil temperature changes, and soil hydraulic balance. We then calibrated and tested our modified RothC (RothC_MM) using the dataset collected in two field mulching experiments, and we used the tested RothC_MM to estimate the expected soil carbon sequestration due to mulching by the year 2050 for the Valencian Community (Spain). Our results show that RothC_MM improved the fit with the experimental data with respect to basic RothC; RothC_MM was able to model the effects of mulch on soil temperature and soil water content and to predict soil organic carbon (SOC) and $CO_2$ observations taken in the field.

**Keywords:** SOC modelling; mulching; conservation agriculture

## 1. Introduction

The sequestration of carbon in the soil has been identified by the IPCC as one of the most readily viable ways to help reach carbon neutrality by 2050, by compensating for $CO_2$ anthropic emission [1]. Carbon (C) is present in soils in inorganic (SIC) and organic (SOC) forms, with agriculture historically depleting SOC reserves, especially in Mediterranean areas, where 20–40% of the original SOC content has been lost after conversion to agriculture [2,3]. This loss of SOC is particularly relevant, considering that agricultural soils occupy about 35% of the global land surface [4]: Lal et al. [5] estimated a global carbon debt due to agriculture of 260 Gt SOC for the top 1 m of soil, with the rate of carbon losses accelerating in the past 200 years [6]. This means that there is a large potential to sequester carbon in the soils through conservation agriculture practices [7], with the added benefit that increasing SOC levels will improve several ecosystem services because of the enhancement in soil structure, fertility, and resistance to erosion [8–10]. This would be extremely important in Mediterranean lands where soil erosion and desertification are major concerns for local farmers and policy makers [11–15]. Therefore, there is a wide

interest in accurately estimating the effects of conservation agriculture practices on SOC trends, at least up to the year 2050.

The effects of agricultural practices on SOC stocks can be estimated using dynamic mathematical models (SOC models) that try to represent the processes that incorporate (input) and mineralize (output) organic compounds in the soil. SOC is the result of the accumulation of decaying organic matter in the soil against mineralization [16]. SOC inputs to the soil come from decaying litter and from the dead remains of roots, lysates, and exudates, i.e., so-called rhizodeposition. SOC outputs consist of carbon oxidized through respiration by soil organisms, as well as lost through erosion and leaching. Models simulate the inclusion of fresh organic matter in the soil as a flux toward different organic carbon pools (2–6 pools, depending on the model). The output of carbon from each pool, including the fresh organic matter pools, is simulated with an exponential decay function, where each pool features a different turnover rate [17,18]. Since it has been widely observed that mineralization rates are modified by soil temperature ($T_s$) and soil water content ($\theta_s$; [19–21]), SOC models include empirical functions to modify the turnover rates of the carbon pools on the basis of both $T_s$ and $\theta_s$ [22].

To obtain reliable estimates of the effects of conservation agriculture practices on SOC stocks for different climates, soil types, and crops, the effects of such practices on soil carbon inputs, $T_s$, and $\theta_s$ should be included fully into SOC models, which can then be run using future climate projections [23]. Among the various SOC models, FAO [24] suggests the use of the less data-demanding RothC, which parametrizes the soil processes mostly with empirical and conceptual functions [17], and the more data-demanding CENTURY, which represents the soil processes with more theoretically sound functions [25].

The less data-demanding RothC is more widely used (due to scarcity of data), but the inclusion of a particular agricultural practice in it requires specific parameters and simple functions to represent the direct and indirect effects of that agricultural practice on the SOC dynamics; reduced tillage, for example, cannot be directly modelled on a less data-demanding model with no soil vertical discretization. Furthermore, less data-demanding models obtain the important $\theta_s$ data from similarly "bucket" soil water budget functions, which are often too simplistic to adequately represent Mediterranean conditions, characterized by long droughts and short, intense precipitation events; for instance, Farina et al. (2013) [26] showed how to modify RothC to improve its performance in Mediterranean climate conditions.

The practice of leaving crop residues or other materials on the soil surface is called "mulching" [27], and it is especially relevant to improving soil and water conservation in orchard inter-rows [28–30]. In addition to improving soil tilth, nutrient availability, and weed control [31], mulching also increases SOC levels [32–34]. Conservation agriculture practices can enlarge SOC stocks directly by transferring carbon to the soil, and/or indirectly by fostering the carbon transfer to the soil from other sources (e.g., plants through rhizodeposition) and/or by decreasing the SOC mineralization rate. The direct effect is the mulch inclusion in the soil either when tilled [35] or carried by soil fauna. The biomass from organic mulches is a matter and energy source for soil organisms, which increases their activity and encourages the binding of organic matter and particles of soil into macro- and micro-aggregates [36], leading to an enhancement in the aggregate stability, restoration of stable C, and, finally, improvement in the SOC content and soil carbon sequestration [37]. The indirect effects are due to changes in soil temperature and water content due to mulch application [38], which can result in changes in root growth and soil ecology and are especially relevant in Mediterranean climates [39–41].

To date, limited attempts have been made to include mulching in SOC models, simulating its effects on soil carbon sequestration by modelling only the direct input of SOC from the biomass of organic mulches [42,43]. Galdos et al. [43] simulated the effect of straw mulch from sugarcane using the CENTURY model; the simulation was focused on understanding whether the straws left on the field contributed directly to the carbon pools and thus to an increase in sugarcane production, leading to a sort of active feedback.

Mondini et al. (2017) [44] modified RothC to include additional exogenous organic matter (EOM) pools and their parameterization by fitting the respiratory curves of 30 soil treatments. Nieto et al. [45] applied RothC to estimate the soil carbon sequestration brought about by mulching in several Mediterranean olive groves. In all these studies, the indirect effects of mulching on $T_s$ and $\theta_s$ were not considered. Moreover, none of the RothC studies on mulch adapted the model to the Mediterranean climate.

The main objectives of this study were to modify the RothC model to estimate changes of SOC due to mulch application in a Mediterranean climate, more specifically in the inter-rows of a Valencian citrus orchard. By modifying RothC, we aimed at including a strategy to sequester atmospheric $CO_2$ in the soil in the RothC model, in order to predict soil carbon sequestration potential at plot and landscape scales [46]. Our specific objectives were to:

- Modify RothC to include the effects of mulch on soil temperature and soil water content, especially in a Valencian climate;
- Calibrate and test the modified RothC using field data, obtained in a three-year-long field experiment [47].

## 2. Material and Methods

To include mulching's direct and indirect effects in RothC, we conducted the following steps: (a) collected the data from two experiments on mulching in citrus orchards in Valencia (Spain) to test the model; (b) built upon the version of RothC developed by Farina et al. [26] (RothC_N) for Mediterranean conditions, in order to introduce the effects of a shallow water table; (c) modified the temperature and soil moisture equations in RothC_N to include the indirect effects of mulching.

*2.1. Modifications of RothC for Mulching's Direct and Indirect Effects in Mediterranean Climate: The RothC_MM Model*

2.1.1. The RothC Model

We decided to use the RothC Rothamsted Carbon model, (RothC-26.3), implemented as described in Coleman and Jenkinson [17]. We selected this model because it requires relatively few input data, it has been tested in a wide range of conditions, and it is already used to estimate SOC stocks in various countries in its standard (Japan [48]; Switzerland, [49]) or modified forms (Australia [50]; United Kingdom [51,52]).

RothC estimates soil water content in terms of monthly topsoil moisture content deficit (TSMD, in mm), based on a "bucket" soil water budget with input coming from precipitation and output from estimated potential evapotranspiration; drainage is simulated by imposing a minimum TSMD corresponding with field capacity, i.e., at a matric potential of $-5$ kPa. The vegetated soil is allowed to dry up to wilting point ($TSMD_{max}$), i.e., at $-1.5$ MPa, and the bare soil to a fraction of it ($0.556\ TSMD_{max}$). A parameter used to modify the SOC pools' degradation rate (*b*, see Equation (1) below) is assumed to change linearly from a value of 1 (no effect on SOC mineralization) at matric potential values of $-0.1$ MPa to a minimal value of 0.2 at wilting point ($-1.5$ MPa). The $TSMD_{max}$ and the TSMD corresponding to $-0.1$ MPa are calculated using an empirical function based on the soil's clay content.

RothC has four active and one inert SOC compartments or pools: Decomposable Plant Material (DPM), Resistant Plant Material (RPM), Microbial Biomass (BIO), Humified Organic Matter (HUM), and Inert Organic Matter (IOM). The monthly organic inputs to the soil are divided into two compartments: DPM and RPM. Then, the output carbon from these two SOC pools flows into BIO and HUM and is lost as $CO_2$. From the BIO and HUM pools, the output carbon again flows into BIO and HUM and is lost as $CO_2$. Each active SOC pool has a different turnover time: DPM (0.165 years), RPM (2.31 years), BIO (1.69 years), HUM (49.5 years). The total output flow of $CO_2$ from the pools can be compared to the monthly average heterotrophic respiration from the soil.

The amount of material that decomposes ($Y_{dec,i}$) in one month in any pool $i$ (where $i$ = {DPM, RPM, BIO, HUM}) is:

$$Y_{dec,i} = Y_i \left( 1 - e^{-abck_i t} \right) \tag{1}$$

where $Y_i$ is the initial quantity of carbon (t C ha$^{-1}$) in a specific pool $i$, $a$ is the rate-modifying factor for temperature, $b$ is the rate-modifying factor for water content, $c$ is the rate-modifying factor for soil cover, $k_i$ is the yearly decomposition rate constant pool $i$, and $t$ is 1/12 [17]. In RothC, weather input data controls temperature and water content modifying factors $a$ and $b$, hence the modelled SOC dynamic in RothC is sensitive to the air temperature and soil water budgets.

2.1.2. Modifications to RothC for Mediterranean Climate: RothC_Med

We adapted RothC to Mediterranean conditions in two steps: first, we implemented the RothC version developed by Farina et al. [26], called "RothC_N"; second, we generalized it to include (a) drainage by macropore flow and (b) shallow water table depth conditions (this generalized version is referred as "RothC_Med").

Farina et al. [26] adapted RothC to semi-arid and Mediterranean conditions working on both the "bucket" model for the calculation of TSMD and on how TSMD translates to modifications of mineralization rates through parameter $b$. The modifications to the "bucket" model included the following: (i) TSMD relationship with soil matric potential and $\theta_s$ based on van Genuchten [53]; (ii) calculation of the van Genuchten parameters using pedotransfer functions [54] (see Supplementary Material S1); (iii) soil is allowed to dry more than wilting point, reaching $-10^3$ MPa; (iv) bare soil is allowed to reach wilting point. The $b$ factor of RothC remains 1 (no effect on mineralization rates) between field capacity and $-0.1$ MPa, then it decreases linearly from 1 to 0.2 between $-0.1$ MPa and wilting point, reaching its maximum effect on mineralization rates [55].

To include the direct and indirect effects of mulching in RothC model, we

- extended the water retention curve function to automatically take in soil water content data obtained from the field, to conduct multi-parameter calibrations of RothC_N;
- introduced a "drainage" empirical parameter (here called "MinTSMD") to represent the effect of macropore flow, e.g., from the formation of cracks; this parameter is a minimum TSMD level which can be calibrated using soil water content time series;
- introduced a sinusoidal function to fit the effect on TSMD of a yearly fluctuating shallow water table; the parameters of this function can be estimated using a local/regional groundwater model or from direct piezometer observations.

All the changes to the equations and algorithm for the calculation of TSMD in RothC to obtain the version "RothC_Med" are detailed in Supplementary Materials.

2.1.3. Modifications to RothC to Include Mulching: RothC_Mulch and Its Combination with RothC_Med (RothC_MM)

To include the direct and indirect effects of mulch in RothC, as observed in the field, we modified the model to account for the following: (i) soil temperature changes, (ii) soil water content regime changes, and (iii) direct C inputs from the mulch material. We called this modified RothC, RothC_Mulch; we gave the name RothC_MM to the model resulting from the combination of RothC_Med and RothC_Mulch. To represent the effect of mulch on soil temperature (either by changes in albedo or thermal barrier), we multiplied the $a$ coefficient in Equation (1) by a parameter (mulch temperature radiation insulation, $M_T$) which can be either directly estimated from $T_s$ data (or estimated using a soil heat transfer model, e.g., Hydrus-1D [56]), in order to reduce the effect of temperature on SOC mineralization:

$$Y_{deg,i} = Y_i \left( 1 - e^{-M_T abck_i t} \right) \tag{2}$$

The inclusion of a parameter that directly multiplies the effect of temperature on SOC degradation is justified by the fact that RothC assumes a linear relationship between parameter *a* and temperature over 10 °C. To represent the effects of mulch on the soil water content regime, the TSMD in the mulch simulations was decreased by a fixed amount (DeltaTSMD mulch parameter), thus indirectly reducing the effect on parameter *b* (Equation (1)). Finally, we introduced the "Mulch C input" parameter to account for the organic carbon inputs contributed by the mulch to the soil either indirectly or directly by the straw, as done previously by Mondini et al. [44] and Nieto et al. [45].

### 2.2. Calibration and Test of the RothC_MM Model

2.2.1. Dataset Used

The model was initialized, calibrated, and tested using a dataset obtained from two experiments on the application of mulch in Valencia [46]. The experiments were carried out in two citrus orchards located, respectively, in the experimental fields of Cajamar ADNAgroFood in Paiporta (coordinates 39°25′2″ N, 0°25′4″ W and altitude of 17 m a.s.l.) and Cooperativa Valenciana del Camp Unió Cristiana in Sueca (coords. 39°12′36″ N, 0°18′23″ W and altitude of 4 m a.s.l.), from March 2019 to January 2022.

According to the Köppen–Geiger classification, the two areas are located in semi-arid hot-summer Mediterranean climate [57]. In Paiporta, the soil is classified as Typic calcixerept and in Sueca as Oxyaquic Xerofluvent according to the Soil Taxonomy [58]. The measured soil parameters were as follows: in Paiporta, clay content 36%, soil depth 20 cm, bulk density of 1.57 kg m$^{-3}$; in Sueca, clay content 32%, soil depth 20 cm, bulk density of 1.4 kg m$^{-3}$.

In each study area (Paiporta and Sueca), there were two control-bare trial plots (Bare) and two straw mulch-covered trial plots (Mulch). The monitored variables were air temperature and relative humidity, with mini-meteorological stations, and rainfall with rain gauges. Close to each weather station, during the first half of March 2019, two capacitance soil moisture and temperature probes (ECH2O 5TM, Decagon Devices, Inc., Pullman, WA, USA) were installed at a depth of 7 cm, one in each inter-row. Soil $CO_2$ emission (soil respiration, $R_s$) was measured once every one-to-two months, starting on March 2019, by fitting static gas-collecting chambers on two permanent-installed PVC collars per trial plot. Every soil $CO_2$ emission measurement lasted 30 min and was conducted in general between 8:00 h and 13:00 h solar time. We minimized the risk of measuring the autotrophic component of respiration by taking soil $CO_2$ measurements as far as possible from the trees and in soils without living grass and roots.

Additionally, batches of soil samples were taken at 20 cm depth using soil augers once every three months between 2019 and 2022. The soil samples were carried to the laboratory, air-dried, ground, and sieved through a 2 mm mesh sieve. SOC was determined by wet oxidation with dichromate according to the Walkley and Black (1934) method [59]. Furthermore, undisturbed soil core samples were taken at 5 cm depth at four points per trial plot to separate the roots and quantify their mass fraction. The C input to the soil from the citrus plants was estimated for both sites and both treatments, based on Mota et al. [60], while the C input from mulch application was based on Dossou-Yovo et al. [35].

2.2.2. Model Initialization

We ran spin-up runs to initialize the carbon pools in RothC_Med and RothC_MM. The carbon pools of RothC are not correlated to measurable quantities; thus, their initial relative relevance (i.e., what percentage of the initial measured SOC is in each pool) are usually estimated with a spin-up run [61]. A spin-up run is a simulation in which the carbon amount in each pool is initially set to zero and then allowed to reach equilibrium under stable yearly weather and soil management conditions. Spin-up runs are crucial to accurately simulate the carbon dynamics in the soil and obtain reliable results, and are widely used in the literature [26,61,62]. The inputs for the spin-up run are the average

weather conditions in the area and the monthly soil carbon inputs from the previous land use. The soil properties are the same as in the main simulations.

We ran spin-up runs for Paiporta and Sueca RothC_Med simulations, since no mulching was applied in the orchard inter-rows before the experiment started. We calculated the "average year weather" inputs using monthly weather data averaged over the years 2009–2020, more specifically for air temperature (°C) and rainfall (mm), while we used the evapotranspiration (mm) estimated by Burguera [63], obtained using the Penman–Monteith method [64]. Data available in the literature shows that citrus orchards were present in the area for at least 40 years before the beginning of the experiment, so the carbon input for the spin-up run was estimated from the literature for Valencian citrus orchards [60].

### 2.2.3. Model Calibration and Test

We applied the Markov chain Monte Carlo method [65] to perform the multi-objective calibration of the models RothC_Med and RothC_MM, based on a dataset obtained from two experiments on the application of mulch in Valencia [47]. The calibration is described as multi-objective because we constrained the model parameters based on three time series: $\theta_s$, $CO_2$ flux from soil, and SOC. The parameters that we calibrated to better fit SOC and $CO_2$ emissions were the input of carbon into the soil (C input), since it was estimated from the literature; and the parameter related to the minimum $\theta_s$ the soil could reach during the dry Mediterranean summer.

To test the model performance, we used the first 22 months of data to perform the calibration, and then we ran the model for another 14 months to check the fit between the observed data and the model predictions for SOC, soil $CO_2$ flux, and TSMD. The comparison between observations and model predictions was performed with a Shapiro–Wilk's test [66] for normality to test the distribution of the difference between measured and predicted SOC values at each site. Moreover, for each dataset, we tested whether the predictions of the RothC_MM model were better than a simple general linear model or not.

The statistical significance of the total difference between the simulated and measured variables was assessed by comparing the RMSE with the value obtained, assuming a deviation corresponding to the 95% confidence interval of the replicated measurements (RMSE95). Moreover, we computed the Akaike information criterion (AIC) score [67] for the three versions of the model.

## 3. Results

### 3.1. Model Initialization

The spin-up simulations took on average 830 years for the SOC pools to reach equilibrium. The relative relevance of each SOC pool, in percentage, for Paiporta and Sueca, respectively, were BIO = 2.18%, 2.19%; DPM = 0.91%, 0.69%; RPM = 13.92%, 13.98% HUM = 82.99%, 83.14%. These fractions were used to derive the initial conditions for our experiment simulations, multiplying them by the SOC stock measured at the beginning of the experiment, which were 28.62 t C ha$^{-1}$ in Paiporta and 36.92 t C ha$^{-1}$ in Sueca.

### 3.2. Model Calibration and Test

The calibration of the water budget part of the modified RothC model resulted in the following values for the new RothC_MM parameters: $-47$ and $-40$ mm for the Paiporta and Sueca "Drought" parameter, respectively; $-20$ mm for both Paiporta and Sueca (Figure 1) "MinTSMD" parameter; 7 mm and 2 months as amplitude and phase shift for the sinusoidal function used only in the Sueca Bare simulation, and zero for all the parameters of the sinusoidal function in all other simulations, since the water table effect on soil water content was negligible in Paiporta and Sueca Mulch.

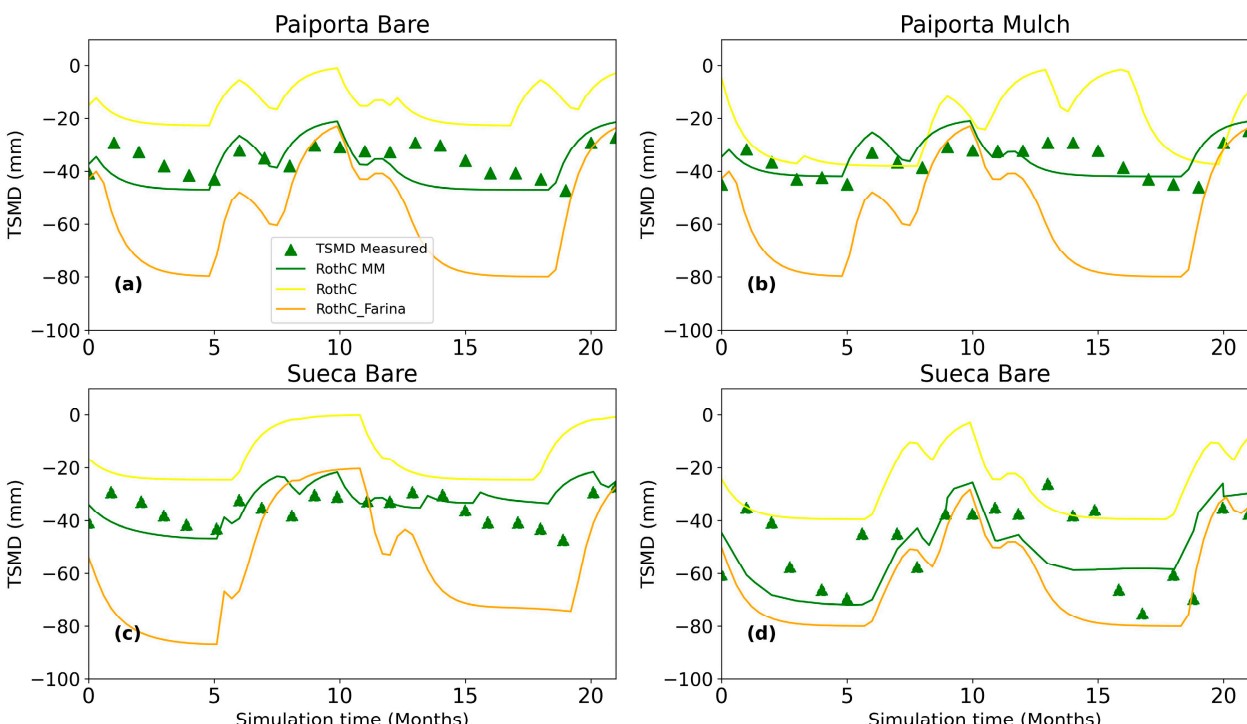

**Figure 1.** TSMD simulated with the basic RothC model, the RothC_N model, and the RothC_MM model after calibration for (**a**) Bare Paiporta, (**b**) Mulch Paiporta, (**c**) Bare Sueca, (**d**) Mulch Sueca; the average soil water content measured in the field, translated to TSMD, is shown for comparison. Simulation time zero corresponds to the beginning month of the experiment (March 2019).

The Bare simulation calibration resulted in a monthly C input set to 0.33 and 0.65 t C ha$^{-1}$ month$^{-1}$ for Paiporta and Sueca, respectively. The calibrated model generally follows the trend of the measured data (Figure 2). The SOC stock simulated by the Bare simulation, and the prediction of the SOC stock for 2 years (from 2019 to 2021), compared with the SOC stock measured, are shown in Figure 2a,b. In Paiporta, the simulated SOC stock is stable in time, while in Sueca, the simulated SOC stock slowly increases. More in detail, the RothC_MM estimates of the SOC stock increase during the period 2019–2021 in the Bare treatments are 2.1 and 4.9 t C ha$^{-1}$ in Paiporta and Sueca, respectively. The simulated soil $CO_2$ emission in the Bare treatments follows the trend of the measurements in both Paiporta and Sueca (Figure 2c,d).

The Mulch simulations calibration resulted in a monthly C input set to 0.61 and 0.63 t C ha$^{1}$ month$^{-1}$ for Paiporta and Sueca, respectively. $T_s$ field observations showed that soil temperature excursion was 3% and 1% lower under mulch treatment in Paiporta and Sueca [47], so the data shows a $M_T$ value of 0.97 and 0.99, respectively.

Soil water content field observations showed that the *Delta TSMD mulch* parameter was −5 mm (as explained in Supplementary Materials). The comparison with the calibrated results from RothC and RothC_MM shows that all models could fit the SOC stock observations and predicted a very similar trend, but basic RothC predicted very different $CO_2$ soil emission trends with respect to RothC_MM for both Bare and Mulch treatments (supplementary Figure S1). To fit the SOC stock and the soil $CO_2$ emission data, however, we had to increase the C input to the soil for RothC and RothC_MM substantially, by the same amount for both Bare and Mulch treatments, up to 0.1 and 0.5 t C ha$^{-1}$ month$^{-1}$ in both Paiporta and Sueca, respectively.

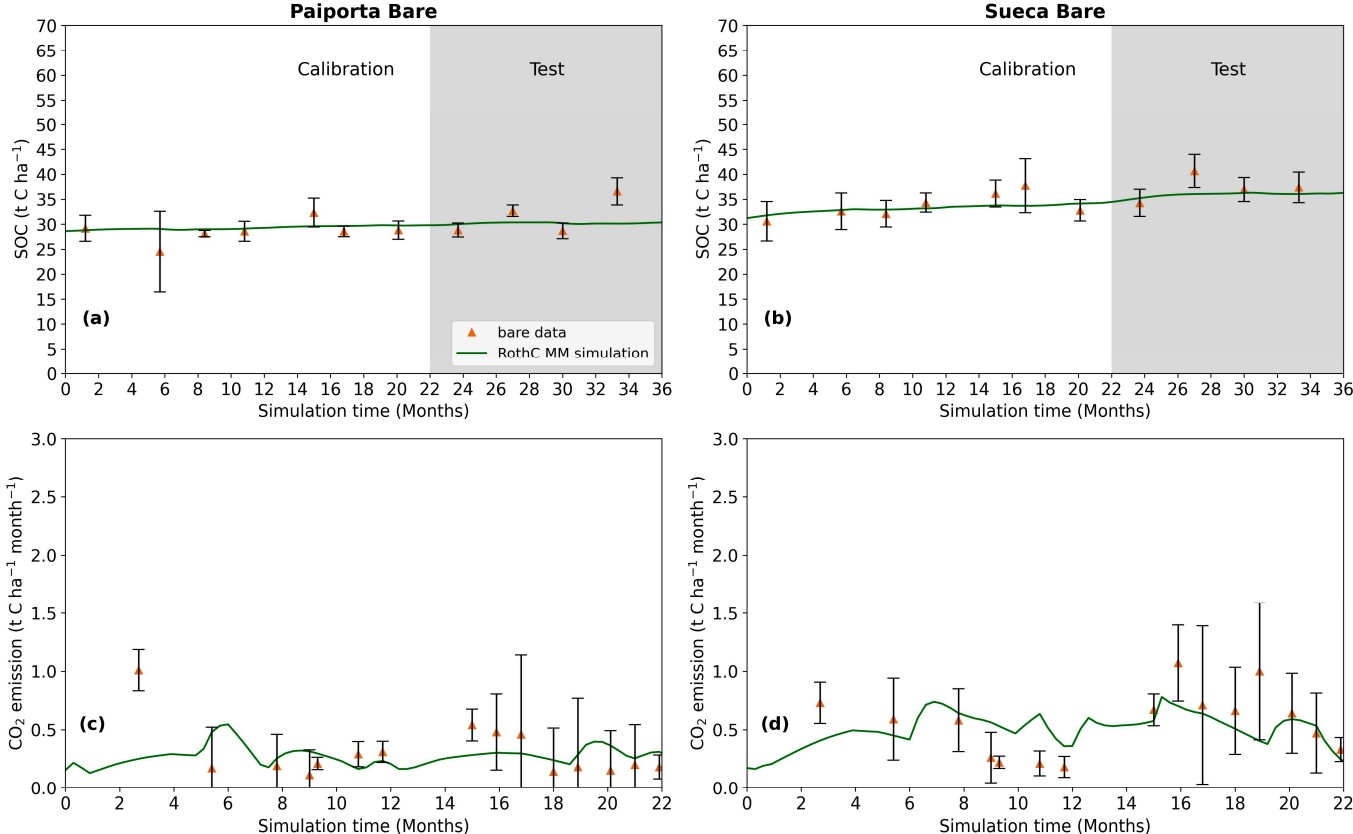

**Figure 2.** Simulated and measured SOC and soil $CO_2$ emission in the Bare simulations after calibration: (**a**) SOC stock in Paiporta, (**b**) SOC stock in Sueca, (**c**) soil $CO_2$ emission in Paiporta, (**d**) soil $CO_2$ emission in Sueca. Simulation time zero corresponds to the beginning month of the experiment (March 2019). The grey areas in (**a**,**b**) corresponds to the test period (year 2021).

Figure 3a,b show the comparison between the SOC stock predicted by the Mulch simulation and the SOC stock measurements for the years 2019–2021. In Paiporta and Sueca, the simulations show a faster increase in SOC stock in the Mulch than in the Bare. The SOC stock predictions for Paiporta fit the measurements, even though SOC stock measurements taken in 2022 show a slight decrease in SOC stock. In Sueca, instead, the SOC stock predictions overestimate the SOC stock observations in the field. More in detail, the RothC_MM predictions of SOC increase, during the period 2019–2021 in the Mulch treatments, are 10.7 and 18.7 t C ha$^{-1}$ in Paiporta and Sueca, respectively. The predicted soil $CO_2$ emission under Mulch follows the trend of the measured soil $CO_2$ emissions in Paiporta (Figure 3c) but tends to underestimate measured soil $CO_2$ emissions in Sueca (Figure 3d).

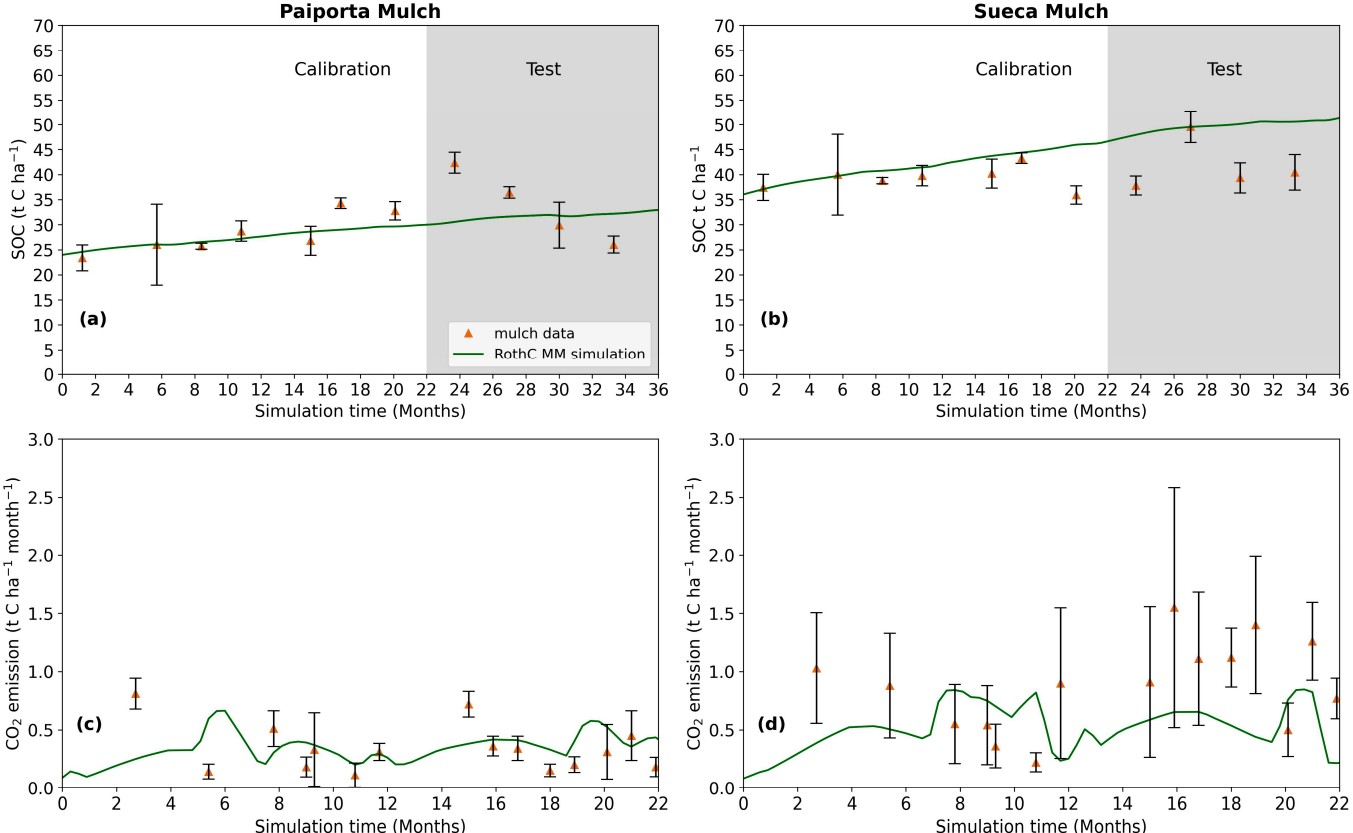

**Figure 3.** Simulated and measured SOC and soil $CO_2$ emission in the Mulch simulations after calibration: (**a**) SOC stock in Paiporta, (**b**) SOC stock in Sueca, (**c**) soil $CO_2$ emission in Paiporta, (**d**) soil $CO_2$ emission in Sueca. Simulation time zero corresponds to the beginning month of the experiment (March 2019). The grey areas in (**a,b**) correspond to the test period (year 2021).

*3.3. Model Test*

The Shapiro–Wilk's test failed to reject the null hypothesis of normality for the difference between the data and model predictions ($p \leq 0.0001$). The relative error of the data RMSE < RMSE95 indicates that the simulated values fall within the 95% confidence interval of the measurements; thus, the model cannot be improved further with these data [68]. The comparison of RothC_MM with the RothC and RothC_N calibrated models for the Bare treatment shows that RothC_MM performed better than the other two versions of RothC, with the RMSE between the observed TSMD values and the simulated one being 21.94, 30.6, and 8.51 for RothC, RothC_N, and RothC_MM, respectively, in Paiporta (Table S3), and 22.19, 29.59, and 8.45 for RothC, RothC_N, and RothC_MM in Sueca (Table S4, Figure 1a,c). The Akaike information criterion (AIC) score [67] for the three versions of the model were as follows: 38.35 and 38.39 for RothC; 41.68 and 41.55 for RothC_N; and 38.65 and 38.39 for RothC_MM, respectively, in Paiporta and Sueca (Tables S5 and S6).

The results obtained for the Mulch treatment simulated using RothC_MM are shown in Figure 1b,d for Paiporta and Sueca, respectively, for comparison. The performance of RothC_MM on SOC stock and $CO_2$ emissions is shown in the Supplementary Materials (Figure S1).

The linear models' fit to the SOC stock data (Section 2.2.3, Supplementary Material, Figure S2) had a slope coefficient of 2.32 in Mulch and 0.67 in Bare in Paiporta (Figure S2a), and 1.41 in Mulch and 1.06 in Bare in Sueca (Figure S2b; a positive slope coefficient indicates an increasing trend in the SOC stock linear model—the larger the slope coefficient, the faster the increase). However, the null hypothesis of non-significant difference could not be rejected, with *p*-value = 0.29 and *p*-value = 0.38 in Paiporta and Sueca, respectively. Thus, the slope coefficients of the linear models for Bare and Mulch are not significantly different.

The RMSE of RothC_N and RothC_MM with respect to the 2021 observations of SOC were 4.86, 2.28, 11.51, and 2.7 t C ha$^{-1}$ for Paiporta Bare, Paiporta Mulch, Sueca Bare, and Sueca Mulch, respectively. The linear models' RMSE with respect to the2021 observations of SOC were 5.18, 2.29, 14.49, and 2.53 t C ha$^{-1}$ for Paiporta Bare, Paiporta Mulch, Sueca Bare, and Sueca Mulch, respectively. The comparison between theRothC_Med and RothC_MM models' RMSE and the linear model RMSE, thus, shows that RothC_N and RothC_MM provide slightly better SOC stock predictions. The Akaike information criterion (AIC) score for the two versions of the model (RothC_N and RothC_MM) are 28.96 and 32.76 for RothC_N, and 33.31 and 37.71 for RothC_MM, respectively, in Paiporta and Sueca.

## 4. Discussion

### 4.1. Calibration and Test Model

The new parameters we introduced in RothC_Med, RothC_Mulch, and RothC_MM are mostly empirical, but are based on well-defined processes, and as such can either be calibrated using time series of $T_s$ and $\theta_s$ measured in the field or estimated independently using a soil model for heat and mass transfer, like Hydrus1D [56]. The use of empirical parameters may be problematic when using RothC_MM when the $T_s$ and $\theta_s$ time series are not available; a physically based description of the processes involved in the mulch effects on soil water content and energy flow would improve the simulations. However, the use of a dedicated model for soil water and temperature, or the inclusion of mulching in a model more data-demanding than RothC (e.g., CENTURY [69]), would increase the amount of parameters and data required to simulate SOC stock dynamics. We believe that our implementation of RothC_MM is flexible enough to accommodate for the availability of data to parametrize the model. Moreover, we would like to point out that RothC is already an empirical model, which needs accurate calibration and testing before being applied to any site.

The RothC_MM model fit the soil water content measured in the field much better than standard RothC or even than RothC_N (Figure 1a,b). RothC_MM was able to predict the low soil water content values reached during the long, dry summer typical of a Mediterranean climate, as well as the lower-than-expected increase in TSMD after a rain event, due to the fact that the intense but short precipitation events in Mediterranean climates often infiltrate only the first few centimetres of soil and then quickly evaporate [70,71], something that cannot be properly simulated when using monthly time steps. RothC_MM was also able to correctly predict the effects of the shallow water table on TSMD levels in Sueca Bare. However, RothC_MM predicted a faster decrease in TSMD levels at the beginning of the dry summer with respect to the observations (Figure 1); this is probably because RothC_MM uses a "bucket" water balance model and does not consider the soil's hydraulic conductivity effect on soil water fluxes.

When comparing simulated and measured soil $CO_2$ emissions, it should be kept in mind that they may not represent exactly the same thing: (i) the simulated soil $CO_2$ emission can be taken as an estimate of soil heterotrophic respiration ($R_h$); the soil respiration was measured as far as possible from the trees, and trenches were dug to check that there were few roots in the soil; (ii) the simulated soil $CO_2$ emission is the estimated monthly average emission of soil $CO_2$ from the soil, while the measurements are taken at a certain time of the day, for half an hour in a single day of the month. The difference between "monthly cumulated" estimated emissions and "measured once per month" emissions are more difficult to analyze, since $R_s$ changes daily and yearly depending on meteorological and biological conditions; however, monthly-only measurements are widespread in the literature [72–75]. Accounting for both differences between simulated and measured soil $CO_2$ emissions, the comparison is aimed only at understanding if the simulated soil $CO_2$ emission is within the range measured in the field.

The calibration procedure resulted in an increase in the modelled C inputs to the soil in a very consistent way over both the Bare and Mulch simulations. This seems to indicate that we initially underestimated the input of organic matter from the citrus trees in the

soil. Our assumption was that mulch straw did not contribute significantly to the input of carbon to the soil; our simulations sustain this assumption, since the calibrated soil carbon input was not higher in the Mulch simulations with respect to the Bare simulations. The use of literature data to estimate the input of organic matter aboveground could partially explain the underestimation with respect to the calibrated values.

The test supported the ability of RothC_MM to predict SOC trends better than both RothC and a linear model. The RMSE and AIC values of the RothC_MM and linear models are on the same order of magnitude; this similarity is likely a result of the short time span of magnitude. The calibration test procedure contributes to the observed convergence in RMSE and AIC values. SOC values change very slowly; trends can require up to 10 years of measurements to be statistically significant [76] and 100–1000 years to reach equilibrium and, thus, to show that their increase is not linear in time. Moreover, the comparison with basic RothC and RothC_Med, calibrated in the same conditions, showed that: (a) using more frequent soil $CO_2$ emission observations, it would be possible to appreciate the improvement brought on by RothC_MM with respect to basic RothC (which does predict very different short-time trends); and (b) even if all models could properly fit the SOC stock data, RothC needed much larger C input in order to do so. The last point means that using RothC to simulate a mulch experiment, without considering the indirect effects of mulching, would result in an overestimation of the C input. We want to stress, however, that our intention in developing the RothC_MM model was not to better fit SOC stock trends only, but to generalize RothC to account for mulch effects and estimate possible trends with different mulch materials.

The results from the calibrated and test models show that the mulch application in both field sites resulted in an increase in SOC stock levels after 3 years with respect to the Bare treatments (compare the trends in Figure 2 with those in Figure 3), even though the SOC stock measurements cannot confirm these results, as shown by the linear model analysis (Section 3.2).

### 4.2. Analysis of Sueca Mulch Simulation

The calibration of the Sueca Mulch simulation was difficult, since it was impossible to fit both SOC stock and soil $CO_2$ emission measurements by changing the monthly C input only. Increasing the monthly C input increases both the simulated SOC stock and soil $CO_2$ emission, and vice versa. However, in the Sueca Mulch treatment, the measurements show a larger soil $CO_2$ emission and a smaller SOC stock value with respect to the non-calibrated RothC_MM simulation; thus, the calibration constrained by both variables results in an overestimation of SOC stock and in an underestimation of the soil $CO_2$ emission (Figure 2c). A better fit could be obtained by identifying a suitable parameter in RothC_MM that, when modified, would result in an opposite change in the two variables, i.e., a parameter that can increase the simulated soil $CO_2$ emission while simultaneously decreasing the simulated SOC stock values.

We identified three possible candidate parameters: *a*, *b*, and $k_i$ in Equation (1), representing the temperature and the soil water content effect on SOC degradation, and the implicit rate of degradation determined by the microbial activity in the soil, respectively. Since all three values are multiplied together in Equation (1), changing one or the other provides the same result, and, thus, we studied them at once by introducing another parameter *j* to reduce or increase the effects on SOC degradation:

$$Y_{deg,i} = Y_i \left( 1 - e^{-j M_T a b c k_i t} \right) \tag{3}$$

The two parameters $M_T$ and *j* are different and have a different meaning, even though both are multiplied in Equation (3) and therefore have the same results on the calculation of $Y_{deg,i}$, just like the parameters *a*, *b*, *c*, or $k_i$ have the same effect on the calculation but a very different physical meaning. $M_T$ is a parameter with a determined value that can be estimated from independent field measurements (Section 2.1.3), while *j* is only used

to study the behaviour of the model and how it could better fit the observations from the Sueca treatment.

As a *gedanken* experiment, we changed *j* between 0.5 and 2, halving and doubling, respectively, the degradation rate of SOC (Figure 4). The results show that doubling the SOC degradation rate substantially improves the fit between the simulated and measured values. We want to stress that the *j* parameter was only used to speculate on the possible causes for the strange behaviour observed in Sueca and was not used in the calculation of the results shown in Section 3.

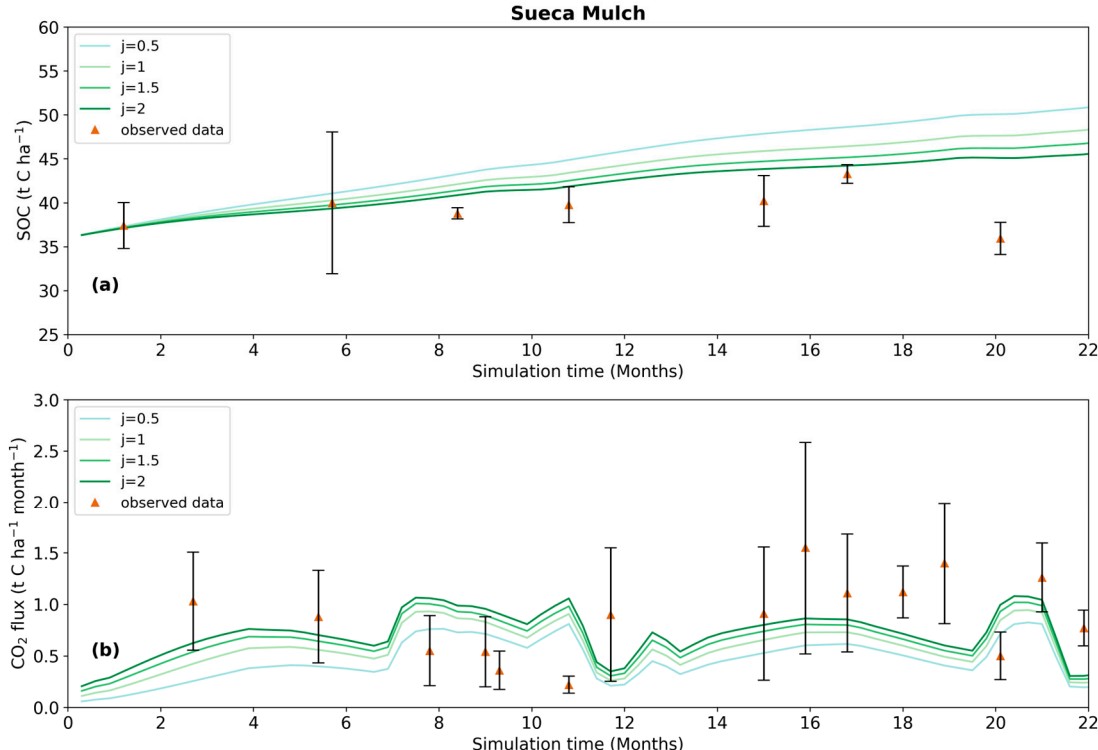

**Figure 4.** Variation in simulated (**a**) SOC pools and (**b**) soil $CO_2$ emission when changing the j coefficient (j = 1 is the baseline simulation) for the Sueca Mulch simulation.

The mulch effects on soil temperature and water content, and consequent effects on SOC degradation rates, have already been included in the $M_T$ parameter and the RothC_MM modifications; as such, *j* can be thought of as changing the $k_i$ parameters, which implicitly represent the microbial community degradation of SOC in the RothC model [16,77]. The parameters $k_i$ represent the intrinsic degradation rate of RothC pools and are seldom modified due to the amount of data required to change them; however, it has been done in some cases [78,79]. A proper calibration of $k_i$ is especially important for the HUM pool, since it is the most important parameter for the determination of long-term SOC stock in RothC [77]. Since $k_i$ is an implicit representation of the degradation activity of the soil biota, to improve its representation in soil carbon cycling models, an explicit representation of the biological and ecological processes in the soil should be considered; various studies are already going in that direction [80–83], but still need further testing before being used to predict SOC changes in field conditions.

### 4.3. 2050 Projections of SOC

We used the modified RothC to assess the effect of mulching on SOC trends up to the year 2050. The projections to the year 2050 are based on the shared socio-economic pathways used in the CMIP6 (Coupled Model Intercomparison Project Phase 6) scenarios representing several temperature increases [84]. To do this, we proceeded as follows: (i) analyzing the weather data (rainfall, air temperature, evapotranspiration) measured in

the years 2009–2020, the same as used in the spin-up run, extracting the monthly average and standard deviation for each variable; (ii) using the monthly average weather data, increasing the temperature linearly in time by 2050, as input for the projection of the SOC trends to the year 2050; (iii) using the monthly standard deviation of each variable to estimate the uncertainty for each variable; and (iv) running a Monte Carlo analysis [85] for a total of 10,000 simulations to perform an uncertainty analysis of the projected SOC stock trends. The scenarios show that, in both study areas, SOC stock does not reach equilibrium even after 30 years of simulation. In Mulch simulations, in 2050 the SOC stock is expected to increase in both areas, even considering the variability of the parameters (Figure 5c,d). In the Bare simulations, instead, the 2050 SOC stock trends are less clear: the uncertainty analysis shows that SOC stock trends could be either increasing or decreasing in time in Paiporta (Figure 5a) or slowly increasing in Sueca (Figure 5b).

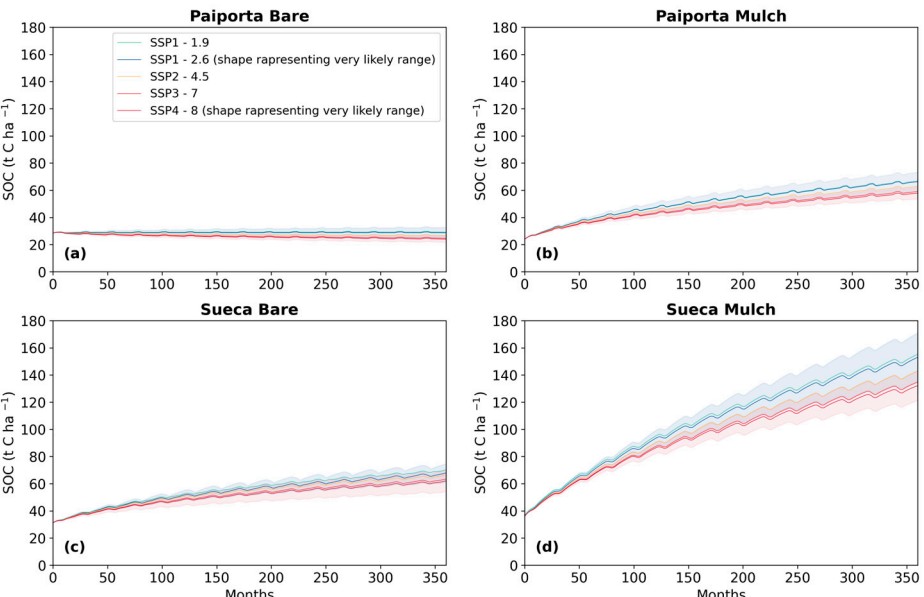

**Figure 5.** SOC projections to year 2050 with CMIP6 scenarios [63], with the related variability shown, as maximum and minimum boundaries, for (**a**) Paiporta Bare, (**b**) Paiporta Mulch, (**c**) Sueca Bare, (**d**) Sueca Mulch. The simulation time starts at 0, being March 2020, the beginning of the experiment, and ends at 360, being March 2050. The weather dataset corresponding to the minimum and maximum SOC stock values simulated by the uncertainty analysis are shown in the Supplementary Materials. SSP stands for shared socio-economic pathway; 1, 2, 3, and 4 are the SSP scenarios; and the last number (1.9, 2.6, 4.5, 7, 8) are the temperature increases, in °C.

Our simulations show an increase in SOC stock of 62.16 t C ha$^{-1}$ by the year 2050, on average. This means that, when mulch application is considered in all citrus orchards in the Valencian Community (159,000 ha), the organic carbon sequestration is expected to reach 9.88 Mt C by the year 2050; however, this assessment should be treated carefully, taking into account first the total production of rice straw in the Valencian Community and then the competition with other uses, e.g., cattle beds. In general, the simulations conducted showed that SOC is expected to increase in all sites apart for Paiporta Bare. A way to improve our simulations would be to decrease the uncertainty related to the future trends for air temperature and rainfall, e.g., using more refined climate change models for the study areas.

### 4.4. Comparison with Other Studies and Study Limitations

Throughout the literature, we found two studies modelling carbon sequestration practices with straw mulch using RothC in the Mediterranean region [44,45]. Nieto et al. [45] simulated the effect of mulch, adding the organic C input of mulch into the model to

compare SOC dynamics in a traditional soil management and in a mulch soil management scenario. Mondini et al. [44] modified RothC, introducing additional pools of decomposable and resistant exogenous organic matter to consider the different nature of several organic amendments, including straw mulch. They conducted laboratory incubation experiments to characterize the exogenous pools of the different amendments.

Since both studies consider only the direct effects of mulching, i.e., the increase in C input to the soil due to incorporation of the mulch biomass, the introduction of mulch in their RothC model has the effect of increasing both the simulated SOC stock and soil $CO_2$ emission from the soil, probably due to priming (see Appendix A). The indirect effects of mulching, however, have the effect of increasing SOC stock, while decreasing the emission of $CO_2$ from the soil. Our RothC_MM model can simulate this indirect effect, which is particularly relevant to soil carbon sequestration practices, since it results in a net gain of carbon in the soil that is not released quickly to the atmosphere as a greenhouse gas ($CO_2$).

A limitation of this study is that the RothC_MM calibration and test was based on data collected over a relatively short period of time (2 years for calibration and 1 year for testing). We acknowledge that the period of the experiment was only three years, even though a similar timeframe has been used in other studies [44,45]. However, we preferred to split the dataset into calibration and test data to check the accuracy and performance of the model, rather than performing calibration only. Since SOC stock changes slowly in time, to perform a proper calibration and test, a longer dataset (around 10 years) would be preferable. For this reason, long-term field studies are suggested (and indeed required) to deliver reliable long-term sequestration projections [86]; some studies are already trying to collect long-term field-collected datasets in standardized platforms to help develop, calibrate, and validate soil carbon models [51,52].

Another limitation is that we did not account for SOC losses due to soil erosion and leaching. We do not believe that soil erosion would affect the study areas analyzed here, since both the Paiporta and Sueca sites are levelled terrain, lower than the surroundings (which also eliminates the issue of rainwater runoff). Leaching, instead, was not measured, and thus was not accounted for in the present study. However, some losses of the dissolvable fraction of SOC are to be expected, due to the soil being in general well drained [47]. Neglecting the loss of SOC due to drainage results in an overestimation of the SOC degradation rate.

Finally, the mineralization process carried out by the soil microbial community is only implicitly represented in RothC, with the constant parameters $k_i$. This potentially limits the validity of the model in soils with different soil microbial communities and raises questions about the source for unexpected fitting problems, as was discussed in Section 4.2. New, ecological soil models are being developed, with the aim of representing the soil microbial community mineralization process explicitly [80–83]; a future study should try to use these novel models to test the hypothesis of the $j$ parameter inserted here.

## 5. Conclusions

Our study shows that it is possible to include both the direct and indirect effects of mulching application on field conditions in a Mediterranean climate:

Our RothC_Med model was able to fit the soil water content observations collected in the field, even in the presence of a shallow water table, with a substantial improvement on both basic RothC and the modification of RothC by Farina et al. (2013) [26].

Our RothC_MM model was able to correctly estimate the SOC stock and soil $CO_2$ emissions for the test dataset. As far as we know, this is the first soil C dynamic model including mulching that was calibrated with SOC stock and soil $CO_2$ emission measurements,

RothC_MM could represent a flexible tool to assess the effect of mulch for small holder farmers in Mediterranean countries, but it requires a long-term field study's implementation to deliver reliable long-term sequestration projections.

For this reason, this study represents an important step toward the assessment of the carbon sequestration potential of mulch. However, some uncertainties about long-term

trends, soil respiration responses to changed conditions (priming effects), and upper ceilings for soil carbon storage call for more long-term experiments, repeated in other places in Mediterranean climates. In conclusion, we emphasize the significance of the indirect effects stemming from sustainable agricultural practices, which influence the mineralization of soil organic carbon (SOC), consequently modulating both the SOC stock and $CO_2$ emissions.

**Supplementary Materials:** The following supporting information can be downloaded at: https: //www.mdpi.com/article/10.3390/soilsystems8010012/s1.

**Author Contributions:** S.P.: Software, Validation, Investigation, Data Curation, Writing—Original Draft, Writing—Review and Editing, Visualization; E.B.: Conceptualization, Software, Validation, Formal analysis, Investigation, Writing—Review and Editing, Visualization; J.M.D.P.: Conceptualization, Methodology, Investigation, Resources, Data Curation, Writing—Review and Editing, Supervision, Project administration, Funding acquisition; D.M.: Methodology, Investigation, Resources, Writing—Review and Editing; F.V.: Conceptualization, Methodology, Investigation, Data Curation, Writing—Review and Editing, Project administration. All authors have read and agreed to the published version of the manuscript.

**Funding:** This research was funded by the project "Evaluación de la fertilidad, secuestro de CO2 y control biológico por la implantación de cubiertas temporales y mulching de paja de arroz en los suelos citrícolas de la Comunidad Valenciana (COVER-CO2)" financed by the European Rural Development Fund (EAFRD) through grant AG_COOP_A/2018/036.

**Institutional Review Board Statement:** Not applicable.

**Informed Consent Statement:** Not applicable.

**Data Availability Statement:** The dataset is available at https://doi.org/10.17632/nmpz7g752t.1.

**Acknowledgments:** We thank the Climate-Kic project, funded by the EU, for financing Simone Pesce's scholarship.

**Conflicts of Interest:** Authors declare no conflicts of interest.

## Appendix A

We want to underline that the predictions in this study are site-specific and account for the site-specific climate and soil characteristics. Figure 4c represents the Sueca control, where the soil quality is generally high due to its soil characteristics as Oxyaquic Xerofluvent, making the soil more resilient to climate change. On the other hand, Figure 4b,d represent the SOC increases due to the mulching practice, which underlines the potential to mitigate or reverse the effect of climate warming, as in the case of Paiporta.

The sensitivity analysis (Figure A1) shows that Mulch C input has a linear effect on SOC stock, as expected, but also controls the relevance of parameter $M_T$, while parameter DeltaTSMD mulch had a negligible effect on SOC stocks. The reason behind this can be elucidated by the fact that, in RothC, SOC pools with slower turnover (HUM and RPM), which are pools with a higher amount of SOC, respond more sensitively to warming than those with a fast turnover (BIO and DPM) [86]. Based on these results, the incorporation of a parameter associated with a specific agricultural practice, such as mulching, which can either enhance or mitigate the temperature effect, yields a significant impact on model simulation that should not be disregarded.

Figure A1a compares the $M_T$ and DeltaTSMD mulch parameters: the former has a considerable effect on SOC content in the RothC, while the latter has little effect (Mulch C input = 0.2). Figure A1b compares $M_T$ and Mulch C input; Mulch C input decreases the $M_T$ effect on SOC increases almost exponentially.

The reason behind this can be elucidated by the fact that, in RothC, SOC pools with slower turnover (HUM and RPM), which are pools with a higher amount of SOC, respond more sensitively to warming than those with a fast turnover (BIO and DPM) [86]. Based on these results, the incorporation of a parameter associated with a specific agricultural

practice, such as mulching, which can either enhance or mitigate the temperature effect, yields a significant impact on model simulation that should not be disregarded.

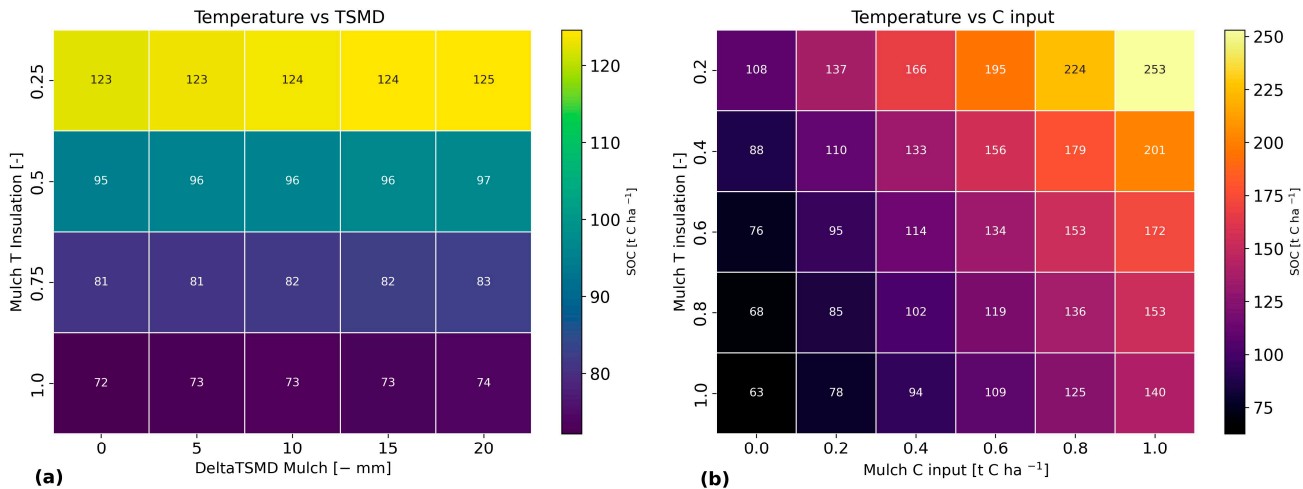

**Figure A1.** Heatmap displaying simulated SOC content. (**a**) The *x*-axis represents DeltaTSMD Mulch values, starting from 0 to −20. The *y*-axis represents Mulch T insulation (MT) values ranging from 0.25 to 1. (**b**) The *x*-axis represents Mulch C Input values, starting from 0 to 1. The *y*-axis represents MT values ranging from 0.2 to 1.

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
