# Peer review of "A Modified Version of RothC to Model the Direct and Indirect Effects of Rice Straw Mulching on Soil Carbon Dynamics, Calibrated in Two Valencian Citrus Orchards"

_soilsystems, doi:10.3390/soilsystems8010012_

Round 1

Reviewer 1 Report

Comments and Suggestions for Authors

Please find the attachment

Comments on the Quality of English Language

Moderate editing of the manuscript is required to correct some of the grammatical and language errors.

Author Response

Dear reviewer, thank you for your help in improving the manuscript. The idea of adding tables was noted, but we fear the manuscript is already heavy with information as it is, so we put the required tables in the supplementary materials. We added some tables in the Supplementary materials that include the values of RMSE, AIC Score, and the calibrated parameters. Moreover we now, in the Introduction section, from line 94 we discuss about of the lack of modelling studies attempt to include the indirect effects of mulching.

All the text changes you required have been implemented. Please check detailed response in attached file.

Reviewer 2 Report

Comments and Suggestions for Authors

The article entitled "A modified version of RothC to model the direct and indirect effects of rice straw mulching on soil carbon dynamics, calibrated in two Valencian citrus orchards" addresses important issues related to the assessment of carbon sequestration and environmental carbon dynamics using the RothC model. This model has been used many times in research on the assessment of soil-based ecosystem services (Abegaz et al. 2022, Jebari et al. 2022, Morais et al. 2018), but, as the authors noticed, the model requires calibration according to the research area. The authors successfully attempted to calibrate the RothC model for orchards in the Mediterranean area (Valencia, Spain) using specific agricultural practices (in this case mulching). In recent years, this practice has become more and more popular, not only in the Mediterranean countries, but also in all climatic zones of Europe. Mulching improves water conditions in soils, helps provide and retain carbon in soils and prevents soil erosion.

Presented research is novel and the manuscript is well prepared, except a few flaws listed below. I recommend a minor revision and acceptance after few corrections.

Line 17: Instead of "destertification; as such, it is an interesting option for" I suggest to change into "desertification. This is why it is a promissing solution for".

Line 18: There is no connection between these two sentences.

Line 28: In CO2 use subscript

Introduction is well written and does not require additional changes.

Lines 117-122: Please rewrite this paragraph. In line 117 instead of ",we:" you can use ". The research was conducted in the following stages".

Material and Methods are sufficiently described and clearly wtitten. Even for poorly experienced in modeling reserchers, the methods used and all the stages will be understandable. There is only one unsignificant correction:

Line 274: Instead of "let the model run" you can make it shorten to "we run the model".

All the applied methods and statistical calculation, according to my knowledge, are commonly used in the modeling reserch. No changes needed.

Results:

Figure 2 was cited before Figure 1.

Line 362: Change 41,68 into 41.68.

Line 379: You missed "c" in comparison.

No further comments. Results were presented and described very well.

Discussion:

Lines 433-438: Divide this sentence into three shorter to increase the readibility.

Line 503: Please change the title of this subchapter

Lines 504-514: Please consider adding this paragraph to the Methodology. I uderstang that the article is multithreaded, but this part should be included in another chapter.

Lines 559-561: Good point.

Conclusions can be imroved. Your research is velauble and you can add here some more arguments and "take home messages" for readers who work on both RothC model and in general, SOC cycle modeling.

Please chack the punctuation and double spaces in the entire manuscript includin the list of references.

Suggested literature to add:

Abegaz A., Ali A., Tamene L., Abera W., Smith J.U., 2022. Modeling long-term attainable soil organic carbon sequestration across the highlands of Ethiopia. Environment, Development and Sustainability, 24, 4. DOI: 10.1007/s10668-021-01653-0

Jebari A., Álvaro-Fuentes J., Pardo G., Batalla I., Martín J.A.R., Del Prado A., 2022. Effect of dairy cattle production systems on sustaining soil organic carbon storage in grasslands of northern Spain. Regional Environmental Change. 22, 2, 67. DOI: 10.1007/s10113-022-01927-x

Morais T.G., Teixeira R.F., Rodrigues N.R., Domingos T., 2018. Characterizing livestock production in Portuguese sown rainfed grasslands: Applying the inverse approach to a process-based model. Sustainability (Switzerland). 10, 12, 4437. DOI: 10.3390/su10124437.

References 41, 56, 80 not found in the manuscript. Reference number 80 cited after 85 for the first time. All the others references were cited correctly.

Author Response

Dear Reviewer, thank you for your help. We fixed all your text-related adjustments. We also included a "take home message" about the importance of indirect effects of agricultural practices in general.

We improved the conclusion, emphasising the importance of indirect effects stemming from the sustainable agricultural practices.
About the comment “Line 504-514: Please consider adding this paragraph to the Methodology. I understand that the article is multithreaded, but this part should be included in another chapter.” We understand the reviewer’s point, but, in our opinion, the speculation about generic parameter j that may represent the effects of mulching on the Sueca soil fauna and microbial community was a discussion point rather than an actual methodology of the study. Since speculations should be kept in the discussion (to ward them off from more grounded analyses in methodology), we decided to keep these line in the Discussion.

More detailed response please check attached file.

Reviewer 3 Report

Comments and Suggestions for Authors

The authors present a relevant devolpment of the widely used RothC model. The text is well written. It reiterates the derivation of some elements of the RothC model. This is a valuable element for users of RothC because the description of the model is rather scattered in the literature.

I recommend only minor revisions

1/ Cite FAO cook book

https://www.fao.org/documents/card/en?details=I8895EN%2f

Citing the FAO cook book that describes the application of RothC may help the reader to appreciate the significance of the paper. The authors may briefly elaborate what on the relevance of their development (if ultimately successful) on the global soil C estimate.

2/ The authors could describe a way-ahead at the end of the paper. Ultimately, the figures show large differences between model and reality and the authors will wish to close in on reality.

Particularly Fig 5 d shows a 3-fold increase in soil C stocks in 30 years. Thats difficult to believe.

3/ Introduction line 33: please clarify: Cseq does not reduce emissions, it compensates for emissions.

Author Response

Dear Reviewer, thank you for your comments. We included all of them in the article; more specifically:

We included the book in the references, and we want to thank the reviewer for the suggestion.

We included a possible way ahead in the conclusions, remarking the importance of long term studies, especially in other areas of the mediterranean, and more in depth analyses of possible “priming effects” and “SOC storage limits”. Moreover, we cite, on line 531, "A way to improve our simulations would be to decrease the uncertainty related to the future trends for air temperature and rainfall, e.g. using more refined climate change models for the study areas. "

We clarified the sentence in the introduction, which was misleading (we talked about “net” emissions, but the term still points to “emissions”, after all). More detailed response please check attached file.
